# The Role of the Sirtuin Family Histone Deacetylases in Acute Myeloid Leukemia—A Promising Road Ahead

**DOI:** 10.3390/cancers17061009

**Published:** 2025-03-17

**Authors:** Piotr Strzałka, Kinga Krawiec, Aneta Wiśnik, Dariusz Jarych, Magdalena Czemerska, Izabela Zawlik, Agnieszka Pluta, Agnieszka Wierzbowska

**Affiliations:** 1Department of Hematology, Medical University of Lodz, 93-510 Lodz, Poland; mellowreine@gmail.com (K.K.);; 2Copernicus Memorial Multi-Specialist Oncology and Trauma Center, 93-510 Lodz, Poland; 3Laboratory of Virology, Institute of Medical Biology, Polish Academy of Sciences, 93-232 Lodz, Poland; djarych@cbm.pan.pl; 4Institute of Medical Sciences, College of Medical Sciences, University of Rzeszow, 35-310 Rzeszow, Poland; 5Laboratory of Molecular Biology, Centre for Innovative Research in Medical and Natural Sciences, College of Medical Sciences, University of Rzeszow, 35-959 Rzeszow, Poland

**Keywords:** acute myeloid leukemia, sirtuins, SIRTs, epigenetics, targeted treatment

## Abstract

The accumulation of mutations in genes responsible for the differentiation, growth, and survival of myeloid precursors plays a major role in the pathogenesis of acute myeloid leukemia (AML). The complex biology of the disease contributes to treatment resistance and a high risk of relapse. Despite the introduction of new drugs in recent years, the prognosis of patients remains unfavorable. Investigating the mechanisms of leukemogenesis and searching for potential therapeutic targets is crucial to improving treatment outcomes. Sirtuins are a family of seven histone deacetylases that are implicated in the regulation of cellular processes in the context of oncogenesis. Increasing our knowledge of their dysfunction in leukemic cells can enhance our understanding of AML. Therefore, this article aims to provide a comprehensive description of the role of sirtuins in AML and bring this issue to the attention of the research community.

## 1. Introduction

Acute myeloid leukemia (AML) is a heterogeneous group of clonal hematopoietic disorders characterized by excessive proliferation and impaired maturation of transformed myeloid blasts. AML is the most common acute leukemia in adults, with a global incidence of 144,645 cases in 2021, and the highest incidence reported in Western Europe (24,638), Canada and the United States (US) (23,676), and East Asia (19,156) [1]. The age-adjusted incidence in the US is 4.2/100,000 per year, accounting for 1% of all cancers. AML prognosis is unfavorable with a 5-year relative survival of 31.9%, according to the National Cancer Institute Surveillance Epidemiology and End Results Program (NCI SEER) [2]. Genetic and epigenetic abnormalities occur in more than 97% of AML patients and accumulate in pre-leukemic hematopoietic stem cells (HSCs). Their multiplicity finally results in the development of abnormal primitive hematopoietic precursor cells called leukemic stem cells (LSCs) and following pathological blasts, leading to the evident leukemic transformation [3,4]. Leukemogenesis is a gradual and heterogeneous process characterized by hyperproliferation of aberrant blasts and blockade of their differentiation into mature myeloid cells [5]. Eventually, this uncontrolled clonal proliferation of non-differentiated myeloid progenitor cells results in the inhibition of normal hematopoiesis in the bone marrow [6].

The mechanism of leukemogenesis includes abnormalities in regulators of apoptosis, tyrosine kinases, and transcription factors such as *TP53* deletions, chromosomal aberrations, or single nucleotide polymorphisms (SNP) of genes participating in key signaling pathways for cellular homeostasis. Moreover, changes in epigenetic modifiers are also critical [7]. This complex pathogenicity results in unsatisfactory AML treatment outcomes, which is related to the inability to eradicate LSCs with conventional chemotherapy and frequent relapses as a consequence [8]. Potential mechanisms of drug resistance in LSCs include quiescence, that is going into a latent state, and exploiting cell cycle regulation to their advantage. In addition, LSCs show lower levels of reactive oxygen species (ROS) and higher expression of autophagy-related genes, which increases toxic molecule elimination. Furthermore, LSCs are rich in altered epigenetic mechanisms, including hypermethylation and histone modification [9]. Finally, they are able to infiltrate the HSC niche, utilize the bone marrow microenvironment for survival, and create a reservoir for residual disease [10]. In addition, there is evidence that the survival of LSCs also depends on oxidative phosphorylation and its critical regulatory metabolite nicotinamide adenine dinucleotide (NAD+) [11]. This implies that targeting AML resistance and eradicating measurable residual disease (MRD) requires a complex therapeutic approach, including the role of epigenetics and cellular metabolism.

Sirtuins (SIRTs) are histone deacetylases, a class of NAD+-dependent proteins that regulate different cell pathways such as oxidative stress, DNA repair, transcriptional regulation, inflammation, and cellular senescence [12]. Thus, their impact on physiological and/or pathological processes is crucial. SIRTs also play a specific and important role in cancer biology [13]. They can both promote and inhibit tumorigenesis, depending on various genetic contexts, tumor type, and stage, by acting as a double-edged weapon [14,15]. Moreover, they affect regulated cell death (RCD), particularly apoptosis, whose dysregulation facilitates the development and progression of AML by permitting the continued survival of cells with activated oncogenes [4].

More recently, SIRTs were discovered to play an important role in the maintenance and differentiation of LSCs, therefore representing a promising area of scientific interest and a potential therapeutic target in hematologic oncology [16,17]. Their role in the pathogenesis, chemoresistance, and prognosis in solid tumors has been widely discussed [18,19,20,21,22,23]. However, there are still no conclusive data evaluating the effect of SIRTs on hematologic malignancies. This knowledge could help to develop new classes of personalized therapies in AML, where, despite novel targeted therapies, treatment resistance remains a challenge. Hence, new therapeutic concepts and tailored approaches to AML treatment are needed in daily clinical practice. We believe it is critical to thoroughly understand the role of SIRTs and their mechanisms of action in AML before effective strategies can be designed.

This review aims to provide a comprehensive overview of SIRTs in AML and, in particular, to examine clinically relevant current and future aspects related to SIRTs, highlighting the molecular background.

## 2. The Sirtuin Family

Since the discovery of the silent information regulator 2 (Sir2) protein in *Saccharomyces cerevisiae* in 1979 [17], seven SIRT homologs (SIRT1-7) with different substrate preferences, enzymatic activities, cellular localization, and targets have been described [24].

One of the main biochemical mechanisms by which SIRTs exert their biological function is deacetylation [25]. To better understand the nature of this process, it is necessary to look at genomics. According to the most recent findings, the human genome contains approximately 19,500 protein-coding genes. At the same time, it is estimated that there may be as many as several hundred thousand to millions of proteins [26,27,28,29]. Three mechanisms are responsible for the diversity of these structures: alternative splicing of precursor mRNA, single amino acid polymorphisms, and post-translational modifications (PTM) [29,30]. One of the principal subtypes of PTMs is acylation, being a process of attachment of an acyl group, in particular an acetyl group (acetylation), to a protein [31]. On the other hand, deacetylation constitutes the dissociation of the acetyl group. Both acetylation and deacetylation are responsible for numerous functions, including an epigenetic mechanism of gene expression control [32].

A specific form of PTM related to the control of gene expression is histone modification. In the 1960s, it was shown that acetylation of the internal lysine residue of a histone causes neutralization of the positive lysine charge, thereby relaxing the chromatin conformation and increasing transcriptional activity [33,34]. Deacetylation, on the other hand, involves hydrolysis of an acetyl group from an acetyl-lysine residue and is carried out by deacetylases, thus leading to chromatin condensation and gene expression silencing [30,35]. Histone deacetylases (HDACs) are important and well-described enzymes [36]. There are 18 known representatives of HDACs divided into four classes. Zinc-dependent classes include classes I (HDACs 1, 2, 3, and 8), IIa (HDACs 4, 5, 7, and 9), IIb (HDACs 6 and 10), and IV (HDAC 11). SIRT1-7 constitute NAD-dependent class III HDACs (Figure 1) [37,38,39].

## 3. Classification of Sirtuins

SIRTs are divided into four classes based on the conserved catalytic core domain (Figure 2). SIRTs 1-3 belong to class I, SIRT4 and SIRT5 to classes II and III, respectively, and class IV is represented by SIRTs 6-7. These proteins also have different intracellular localization. SIRT1, 6, and 7 are mainly located in the cell nucleus; SIRTs 3–5 are mitochondrial, whereas SIRT2 is cytosolic and can move into the nucleus during mitosis [12,40]. SIRTs are not only histone deacetylases. They can target proteins regulating cell apoptosis and proliferation and modulate the activity of both suppressor proteins (p53) and protooncogenes (cellular myelocytomatosis oncogene; c-MYC), as well as enzymes involved in metabolic processes (carbamoyl phosphate synthetase 1; CPS1, glutamate dehydrogenase; GDH) [41]. SIRTs 1-3 mainly exhibit strong deacetylating properties, while the other members of the SIRT family are weaker deacetylases but present additional catalytic functions, like demalonylation, deglutarylation, and desuccinylation. Although SIRTs are best known for their deacetylating activity, some of them have unique features; for instance, SIRT4 acts rather as an adenosine diphosphate (ADP) ribosyltransferase, while SIRT6 also demonstrates ADP-ribosylation capacity, through which it regulates the poly (ADP-ribose) polymerase 1 (PARP1) protein [12]. Different SIRTs were also shown to act oppositely at a given point of capture, as well as depending on the biological context, especially concerning oncogenesis, the mechanism of which remains poorly understood [42]. Indeed, the biology and microenvironment of each tumor differ. Therefore, the description of SIRTs’ action in AML requires special attention and is certainly an interesting topic for exploration.

## 4. Overview of the Role of the Sirtuin Family in AML

### 4.1. Sirtuin 1

SIRT1 is the oldest known and best-described human SIRT. Its role in the context of cancerogenesis and its impact on pathways controlling apoptosis and proliferation are widely discussed. This protein has been considered oncogenic since its deacetylating effect on p53 with the consequent silencing of its function was described, for example, in lung cancer [43,44]. However, in subsequent years, the tumor-suppressive properties of SIRT1 were demonstrated, depending on the type of neoplasm [45]. To date, several potential mechanisms through which SIRT1 may influence the development and course of AML have been described.

The role of SIRT1 in AML may be clearly associated with early stages of the disease. LSCs are thought to be precursors of leukemic blasts and are considered the most possible source of relapse [46]. They are rich in the CD34+CD38− subset, and it was proven that SIRT1 expression is increased in these cells. It does not explain their effect on leukemogenesis, however, sheds light on their potential involvement in the formation or survival of LCSs [47]. SIRT1 is upregulated in cells derived from AML patients classified into the intermediate- or high-risk group compared to the low-risk group, according to the National Cancer Center Network (NCCN) classification [47,48]. Higher expression of SIRT1 was also found in samples obtained from AML patients with an internal tandem duplication in the FMS-like tyrosine kinase 3 gene (*FLT3-ITD*) [47,49], which is one of the most common AML subtypes (approximately 25% of all cases) and is connected with an increased risk of relapse [50,51]. SIRT1 overexpression is related to the c-Myc network, which reduces SIRT1 ubiquitination and enhances its stability [47]. Moreover, Li et al. found that SIRT1 expression correlates with ubiquitin-specific peptidase 22 (USP22) expression and, through the c-Myc/USP22/SIRT1 pathway, reduces p53 activity and thereby promotes LSC survival [47]. Elevated levels of SIRT1 were described not only in LSCs but also in blasts derived from AML patients compared to healthy controls [52]. A study by Li-Cheng et al. demonstrated that both gene and protein expression levels of SIRT1 and peroxisome proliferator–activated receptor gamma coactivator (PGC)-1 alpha were higher in AML cells resistant to cytosine arabinoside (AraC) than in primary AML samples [53]. Interestingly, Tian et al. described a negative correlation of *SIRT1* gene expression with interferon-regulatory family 9 (IRF9), a protein that can inhibit SIRT1 and thus promote p53 activation [54]. Yet, another important aspect is the potential of epigenetic regulation of gene expression by SIRT1. Chen et al. demonstrated its role in regulating chromatin status in histone-lysine N-methyltransferase 2A (*KMT2A*) mutated leukemias in which SIRT1, acting as a disruptor of telomeric silencing 1-like (DOT1L) antagonist, can induce gene silencing [55].

Recently, another pathway related to SIRT1 and associated with the forkhead box protein 1 (FOXP1) has been described. The FOXP1 regulator affects the survival of mouse hematopoietic stem cells by suppressing oxidative stress. It has also been shown to affect the growth of AML cells. SIRT1, on the other hand, serves as a sensor of oxidative stress in AML cells. Induction of FOXP1 overexpression increases SIRT1 expression by enhancing SIRT1 protein stability. Consequently, this contributes to the resistance to chemotherapy of AML cells [56]. SIRT1 effects on forkhead box protein O1 (FOXO1), signal transducer and activator of transcription 5 (STAT5)—related pathways, and tyrosine kinases like BCR-ABL, promoting leukemogenesis in chronic myeloid leukemia (CML), have also been described [57,58].

It has been shown that the expression and activity of SIRTs can also be influenced by micro RNAs (miRNAs, miRs). miRNAs are short, single-stranded, non-coding RNAs consisting of ~22 nucleotides. By cleavage of complementary mRNAs or their destabilization, they are involved in the post-transcriptional regulation of gene expression, including *SIRT* genes [59]. For example, it has been proven that miR-9 directly binds to *SIRT1* mRNA 3′ and silences its translation process. In AML, Zhou et al. showed that patients with the t(8;21) translocation, involving the *RUNX1-RUNX1T1* gene fusion, have lower miR-9 expression and higher SIRT1 expression [60].

In conclusion, numerous reports indicate an oncogenic effect of SIRT1 in hematological malignancies, including AML. By affecting p53 protein and c-Myc-related signaling pathways, it indirectly inhibits apoptosis, promotes maintenance of LSCs, and leads to survival of AML blasts.

### 4.2. Sirtuin 2

SIRT2 is the second representative of the SIRT family. It is known as a cytoplasmic SIRT but has the ability to enter the nucleus during mitosis, where it affects chromatin condensation through interaction with histone H4 Lys 16 [61]. SIRT2 is highly expressed mainly in the cytoplasm of neurons and oligodendrocytes in the brain and thus is considered to be involved in neurological disorders [62]. SIRT2 has pleiotropic activity in tumorigenesis and can act as a tumor promoter or suppressor depending on factors like microenvironment or metabolic requirements of particular cells [63,64]. SIRT2 might also impact tumors indirectly by affecting immunologic processes through T-cell metabolism regulation [65]. Published data showed that SIRT2 inhibits not only the humoral response but also macrophages, myeloid-derived cells, and tumor-associated neutrophils in tumor tissues [64]. Interestingly, it can increase lactate dehydrogenase (LDH) activity through its deacetylation, which affects lactic acid accumulation and promotes tumor cell proliferation [66]. There are data that inhibition of SIRT2 can induce apoptosis and autophagy in leukemic cell lines, including human T-cell lymphotropic virus type 1 (HTLV-1) transformed T-cells [67]. These findings indicate the role of SIRT2 in the pathophysiology of AML.

Both *SIRT2* gene and protein expression were found to be elevated in AML cells compared to normal tissues or CD34+ cells from healthy individuals [64,68]. Moreover, Deng et al. showed its significantly higher expression in bone marrow samples from patients with high- or intermediate-risk AML than in low-risk AML according to the NCCN. High *SIRT2* expression was associated with shorter overall survival (OS) and event-free survival (EFS). *SIRT2* is presumed to be related to genes of the mitogen-activated protein kinase (*MAPK*) and vascular endothelial growth factor (*VEGF*) signaling pathways, the upregulation of which correlated with SIRT2 expression [69].

Another mechanism of SIRT2 action has been noted in *KMT2A*-mutated AML. The MLL1 protein, encoded by the *KMT2A* gene, is a methyltransferase that regulates hematopoiesis. This process is mediated by modification of histones in the promoters of target genes. The presence of an abnormal form of the protein leads to neoplastic transformation of hematopoietic cells [70,71]. Hao et al. demonstrated that SIRT2 deacetylase acts as an upregulator for poor prognostic *KMT2A-ENL* fusion, and loss of SIRT2 increases the chemosensitivity of AML cells bearing this rearrangement [72].

Moreover, the upregulation of *SIRT2* mRNA and protein expression in bone marrow samples was observed in patients with relapsed AML as compared to the newly diagnosed population. In addition, SIRT2 inhibition resulted in decreased expression of the multidrug resistance-associated protein 1 (MRP1), which is considered to be a chemotherapy resistance factor in leukemic cells. A positive correlation between SIRT2 expression, extracellular signal-regulated kinase 1 and 2 (ERK1, ERK2), and the anti-apoptotic B-cell lymphoma 2 protein (BCL-2) was also observed. On the other hand, SIRT2 expression correlated negatively with wild-type p53 levels. These results indicate that the ERK1/2 signaling pathway is related to SIRT2, and this interaction may affect multidrug resistance to chemotherapy, as well as modulate apoptosis by regulating both pro- and anti-apoptotic proteins [73].

SIRTs make their mark, as already stated, not only by modifying histones or affecting p53. The mammalian target of rapamycin (mTOR) is a phosphorylase that can stimulate tumor growth by acting on proteins that control cell division and proliferation. It is part of the phosphoinositide-3-kinase/protein kinase B/mTOR (PI3K/AKT/mTOR) signaling pathway. Due to its importance in tumorigenesis, inhibition of this pathway is used for tumor suppression [74]. Luo et al. have recently demonstrated that the PI3K/mTOR inhibitor impacted SIRT2-dependent pathways. Moreover, they showed that inclusion of the SIRT2 inhibitor had a synergistic suppressing effect on tumor growth and increased survival in a murine model. An increase in wild-type p53 expression was observed when both PI3K/mTOR and SIRT2 inhibitors were used [75]. Furthermore, Dan et al. assessed the AKT/glycogen synthase kinase-3 beta (GSK3β)/β-catenin signaling pathway in AML for the effects of nicotinamide phosphoribosyltransferase (NAMPT) and SIRT2, which showed high expression in blasts. The authors demonstrated that NAMPT and SIRT2 affect the survival and proliferation of blastic cells potentially through inhibition of the mentioned signaling pathway [68]. In another study, the researchers showed that NAMPT and SIRT2 are involved in leukemic transformation through activation of GSK3β, leading to accumulation of the β-catenin oncogene in the cell nucleus [76].

Abnormal proliferation, being as aggressive as that observed in acute leukemia, requires complex anabolic processes. These are mediated, inter alia, by the pentose-phosphate pathway, the oxidative branch of which is glucose-6-phosphate dehydrogenase (G6PD). It plays an important role in AML cell proliferation, and its increased activity and downregulated acetylation were proved in AML blasts compared to normal control. SIRT2, by deacetylating G6PD, promotes NADPH production, which drives AML clone proliferation and development [77].

Considering natural agents that have the potential to modulate SIRTs, Russo et al. investigated the effect of flavonoids on the monocytic leukemia THP-1 cell line. Naringenin and hesperetin decreased *SIRT2* gene expression and thus *TP53* deacetylation while modulating the expression of the regulatory proteins p21 and cyclin E1. In this study, flavonoids exerted an antiproliferative effect on leukemic cells by potentially acting on a signaling pathway of which SIRT2 is an essential point [78].

Apart from the mechanisms affecting blast proliferation and survival, clinical management of side effects and toxicity of AML treatment are other important aspects. Induction of remission in acute leukemia, contributed by anthracycline therapy, certainly has a strong toxic effect on the myocardium. Zhao et al. demonstrated that miRNA-140-5p exacerbates this toxicity by increasing oxidative stress through nuclear factor erythroid 2-related factor 2 (NRF2) and SIRT2 targeting. Blocking the miR-140-5p/SIRT2 axis may facilitate treatment of cardiotoxicity during anthracycline chemotherapy [79]. miR-145 is another miRNA, whose effect on SIRT2 in AML may be worth further investigation. Its expression positively correlates with proapoptotic genes, suggesting its potential role in the induction of apoptosis in AML. SIRT2 has appeared to demonstrate a significantly negative correlation with miR-145, and this observation may imply that SIRT2 exhibits pro-leukemic activity [80].

### 4.3. Sirtuin 3

SIRT3 is a mitochondrial SIRT, involved in the metabolism and maintenance of mitochondrial homeostasis by regulating the acetylation of other proteins. Research shows that SIRT3 influences aging processes. Its association with longevity and delay in degeneration of neurons or the macula has been widely described [81]. Additionally, SIRT3 plays a role in both oncogenesis and tumor-suppressor processes [81,82]. In the pathway with phosphatase and tensin homologue deleted on chromosome 10 (PTEN) and mouse double minute 2 homolog (MDM2), it can indirectly inhibit p53 degradation. In consequence, glycolysis in wild-type *TP53* cancer cells is inhibited, which leads to tumor suppression [83]. Furthermore, SIRT3 can deacetylate and simultaneously deactivate cyclophilin D and lead to inhibition of glycolysis in breast cancer cells [84]. On the other hand, it can enhance the activity of isocitrate dehydrogenase 2 (IDH2), resulting in oncogenesis stimulation, especially in hematological malignancies [85].

SIRT3 has been confirmed to demonstrate a promoting effect on human LSCs by regulating oxidation of fatty acids required for the production of adenosine triphosphate (ATP). O’Brien et al. demonstrated that SIRT3 inhibition by YC8-02 reduces fatty acid oxidation activity and substrate availability for the tricarboxylic acid cycle, thereby affecting oxidative phosphorylation, decreasing ATP production, and leading to LSC death [86]. Furthermore, it has been shown that disruption of cholesterol homeostasis sensitizes LSCs to SIRT3 inhibitors and enhances their apoptosis. Interestingly, inhibition of SIRT3 sensitizes LSCs to venetoclax, which acts as a specific inhibitor of BCL-2 protein [87]. In addition, SIRT3 was shown to inhibit chemotherapy-induced mitochondrial ROS production and increase oxidative phosphorylation in AraC-treated and untreated AML cells. Inhibition of SIRT3 led to blast destruction synergistic with Ara-C both in vitro and in mouse models [88]. The above mechanisms indicate the AML-promoting activity of SIRT3 as well as its impact on chemoresistance, both to classical chemotherapy and to targeted drug therapy.

Another mechanism, affected by SIRT3 in AML, is a specific type of post-translational modification of proteins, called sumoylation. This is a covalent attachment of small ubiquitin-like modifier (SUMO) proteins to the target protein [89]. Dysregulation of SIRT3 sumoylation may enhance AML blast chemoresistance. Desumoylation of SIRT3 increases SIRT3 activity and inhibits the hairy and enhancer of split-1 (HES1) transcription factor, being a part of the Notch1 signaling pathway. As a result, fatty acid oxidation increases, which enhances resistance to chemotherapy. In contrast, inhibition of desumoylation or overexpression of HES1 acts synergistically with AraC in eliminating leukemic blasts in vitro and in mouse models [90]. Jiao MA et al. also demonstrated the importance of SIRT3 sumoylation in AML cell survival. They pointed out that SIRT3 desumoylation decreases the generation of ROS and alters mitochondrial metabolism, which may consequently contribute to AML resistance to cytostatic agents [91]. Other studies reveal that downregulation of SIRT3 sensitizes AML blasts to conventional chemotherapy [88].

However, in our study, *SIRT3* appeared to be downregulated in patients with secondary AML, whose prognosis is generally unfavorable [92]. Similarly, some reports demonstrate relatively low expression of SIRT3 in AML, depending on the disease subtype [23]. This may indicate a bidirectional action of this type of SIRT in oncogenesis; therefore, these mechanisms certainly require further research.

### 4.4. Sirtuin 4

SIRT4 is the second representative of the SIRT family that acts mainly in mitochondria [93]. SIRT4 is known to act as both ADP-ribosyl-transferase and lysine deacetylase and plays a role in regulating metabolism, lipid homeostasis, insulin secretion, and the cell cycle [94]. It is particularly prevalent in white blood cells and adult thymus [95]. SIRT4 is considered a tumor suppressor gene. Its overexpression was shown to inhibit glutamine metabolism and thus the growth of Burkitt’s lymphoma cells [96,97]. As compared to healthy tissues, decreased mRNA levels of *SIRT4* were observed in many neoplasms, including breast, stomach, thyroid, bladder, colon, and ovary cancers [98]. Reduced SIRT4 expression was also associated with advanced pathological grading and poor survival in many types of solid tumors [20,94]. Other studies, however, provide evidence for esophageal cancer-promoting SIRT4 activity [99].

There is little data regarding SIRT4’s impact on AML. Bradbury et al. performed a study characterizing the expression of all 18 histone deacetylases in AML blasts compared to peripheral blood mononuclear cells (PBMC) as well as CD34+ progenitors derived from the umbilical cord or granulocyte colony-stimulating factor (G-CSF)-stimulated healthy donors. SIRT4 and HDAC5 were the only genes consistently underexpressed in leukemic blasts in comparison with PBMC and CD34+ progenitors. What is more, HDACs inhibitors (HDIs) induced diverse changes in HDAC gene expression, depending on the inhibitor, its concentration, or cell type. However, only SIRT4 of all the SIRT family showed consistent upregulation with all inhibitors. It may imply that it demonstrates opposite activity compared to other SIRT members under the conditions of HDI implementation [52].

In our previous studies, we also noted a strong positive correlation between *SIRT4* and anti-oncogene wild-type *TP53* expression in leukemic cells [100].

There is evidence that SIRT4 might influence the pathogenesis of other hematologic malignancies. Knocking out SIRT4 in mice led to spontaneous development of lymphomas [96]. Moreover, it was found that SIRT4 can inhibit the growth of Myc-induced B-cell lymphoma [97]. The presented data highlight the tumor-suppressing effect of SIRT4 in the context of neoplasms originating from blood and bone marrow cells.

On the other hand, some reports reveal an upregulation and promoting effect of SIRT4 on the formation of malignant tumors, including AML. O’Brien et al. showed that SIRT4 knockdown had an inhibitory effect on the viability and colony-forming ability of AML specimens [87]. One source points to overexpression of SIRT4 in AML, but at the same time it questions its direct causative relationship with leukemogenesis. High expression of SIRT4 may be explained by boosted proliferative activity of blasts and thus increased cells’ demand for mitochondrial metabolism, which can lead to SIRT4 upregulation [101].

The above data, as well as literature describing SIRT4 in oncology generally, demonstrate that SIRT4 is predominantly established as a tumor suppressor [102]. This molecule plays an important role in the damage response of mitochondrial metabolism, which may inhibit tumor initiation [96]. SIRT4 is believed to play a less important role in oncogenesis, and there is little data on this relationship. Yet, as it happens with the entire SIRT family, the mechanisms are ambivalent, and pro-oncogenic actions of SIRT4 have also been described. Jeong et al. claim that one of the explanations for this phenomenon is the fact that cancer cells are able to modulate the stress response in their favor, and this ability enables them to grow [103].

### 4.5. Sirtuin 5

SIRT5 is another mitochondrial SIRT [104]. It is a lysine deacetylase, but some data confirm its demalonylase, deglutarylase, and desuccinylase activity [105,106,107]. SIRT5 acts mainly in the intermembrane spaces of mitochondria, and its expression is highest in the heart, brain, liver tissues, and lymphoblasts [108]. It regulates a wide range of metabolic pathways, including those related to energy metabolism and apoptosis [109]. SIRT5 plays an important role in the downregulation of oxidative stress and can protect some cancer cells from stress-induced apoptosis [110,111]. It may also increase the expression of BCL-2 and decrease apoptotic bcl-2-like protein 4 (BAX) [112]. These proteins are involved in signaling pathways crucial for cell survival, and a common mechanism for cancer cells to avoid apoptosis is the upregulation of BCL-2 and loss of BAK. BCL-2 is also a major target for venetoclax-based targeted therapies in hematologic malignancies [113]. These data may suggest SIRT5′s pro-oncogenic activity, possibly leading to tumor maintenance.

SIRT5 is required to make mitochondrial oxidative phosphorylation and other key processes run in AML blasts [114]. Yan D. et al. described the dependence of survival and growth of primary AML cells on SIRT5, being a regulator of energy metabolism. SIRT5 can be efficiently inhibited by the NRD167 molecule, which increases mitochondrial superoxide, reduces oxidative phosphorylation, and induces apoptosis. It was also confirmed that the knockdown of SIRT5 in AML blasts reduced colony formation and inhibited growth in AML cell lines [115]. Such an effect was not observed in normal cells, which suggests that SIRT5 knockdown in AML cell lines preferentially targets AML blasts [116].

It is supposed that SIRT modulation could be a therapeutic strategy in AML. Rajabi N. et al. designed a masked tetrazolium-containing molecule showing selective cytotoxicity and enhanced activity against SIRT5-dependent AML cell lines. The mechanism is based on the inclusion of carboxylic acid isosters, which play a role in binding to the active site of the enzyme [117]. Other data confirm the dependence of AML cells on SIRT5, but this correlation was not found for CD34+ cells. What is more, SIRT5 knockout mice demonstrated impaired transformation of hematopoietic cells by myeloid oncogenes in vitro, and attenuated leukemogenesis was observed in vivo [118].

Wang M. et al. proved synergistic activity between the administration of venetoclax and inhibition of SIRT5 in AML. Venetoclax-resistant patients had elevated NAD+ metabolism, and the authors hypothesized that knockdown of SIRT5 may increase venetoclax activity via attenuation of NAD+ metabolism. Indeed, this combination leads to numerous processes, like increased production of ROS, decreased ATP generation, and consequently increased apoptosis in AML cells [119]. Moreover, Zhang J. et al. investigated SIRT5 effects on the AML cell lines and concluded that SIRT5 is involved in mediating AML development through glycine decarboxylase (GLDC) succinylation [120].

Some reports indicate that deacetylase and desuccinase activities of SIRT5 can be modulated by resveratrol, a natural polyphenol characterized by antioxidant and anticancer properties [121]. Ozkan T. et al. found that in K562 CML cells, resveratrol decreased the SIRT5 level and induced apoptosis, which, hypothetically, may have been mediated by the SIRT5-lowering effect [122]. However, there are no data regarding the effect of resveratrol on AML.

Being a key metabolic regulator in AML, SIRT5 is believed to be essential for the survival of LSCs. Knockout of SIRT5 in mouse models led to rapid apoptosis of AML cells and disease regression. However, some cell lines were resistant to SIRT5 inhibition. These findings were analyzed by Bateman B. et al., and their study revealed that AML samples with nucleophosmin 1 mutations (*NPM1^mut^*) showed predominantly SIRT5-independent nature. It has been shown that cells with *NPM1^mut^* increased mitochondrial DNA in the cytosol. Proliferation of SIRT5-independent *NPM1^mut^* cells results from adaptation to mitochondrial dysfunction. In contrast, SIRT5 depletion in SIRT5-dependent samples showed apoptotic effects on AML cells [123]. This demonstrates the complex nature of AML, which depends on genetic aberrations, including changes that cause mitochondrial damage. On the other hand, it has been observed that glioma cells with *IDH1* (R132H) mutations (*IDH1^mut^)* exhibit hypersuccinylation leading to increased expression of the antiapoptotic BCL-2. *IDH1^mut^* enhances succinylation, shutting down mitochondrial respiration, and thus inducing apoptosis blockade. Overexpression of the SIRT5 desuccinylase results in inhibition of tumor growth in these cases [124,125]. In contrast, SIRT5 action in the *IDH1^mut^* AML subgroup still has not been thoroughly described.

The vast majority of data on the effect of SIRT5 on AML point to its leukemogenesis-promoting nature. This observation is not fully supported by the literature concerning solid tumors in general, where its mechanisms are described as either tumor promoting or inhibiting, even in the same tumor subtype, depending on the biological context [126]. However, given the heterogeneity of AML, analyses of the effect of SIRT5 on molecular subgroups, such as those with *IDH* or *FLT3* mutations, are still limited.

### 4.6. Sirtuin 6

SIRT6 is a nuclear SIRT, which acts as both a histone deacetylase and ADP-ribosylotransferase [127]. Its presence is mainly observed in skeletal muscles, liver, brain, heart, and thymus [128]. It is also known to regulate processes like DNA repair, metabolism, inflammation, and telomere maintenance [127,129]. SIRT6 is called the longevity gene, as it plays a key role in aging inhibition [130]. However, its ability to repair DNA also makes it widely discussed in the context of oncogenesis. The capacity of SIRT6 to inhibit tumor formation consists in maintaining genome integrity, and downregulation of SIRT6 in ovarian, breast, colon, and liver cancers has been documented [131,132,133,134]. Furthermore, according to the Cancer Cell Line Encyclopedia report, more than one-third of cancer cell lines included in their database contain a deletion of the *SIRT6* gene [135]. There are data supporting an anti-oncogenic effect of SIRT6. However, this protein, like other SIRTs, also exhibits negative characteristics; its increased expression has been shown in squamous cell carcinoma and hematological malignancies, including AML [136,137,138,139,140].

Cagnetta A. et al. observed that SIRT6 was overexpressed in AML CD34+ blasts but underexpressed in normal CD34+ hematopoietic stem cells. They also demonstrated that depletion of SIRT6 in AML cells increases their susceptibility to classical cytotoxic drugs such as daunorubicin and AraC. The authors hypothesize that the DNA repair functions of SIRT6, which are beneficial in the physiological state, work in favor of AML here, and SIRT6 inactivation increases the toxicity of chemotherapy [140]. Furthermore, SIRT6 is also referred to as a guardian of the genome of AML cells, which may be responsible for chemoresistance. An Italian study conducted by Cea M. et al. confirmed high expression of *SIRT6* in 200 AML patients, as well as a positive correlation between *SIRT6* expression and high chromosome instability, and proved its adverse prognostic value. SIRT6 upregulation was also observed in AML cell lines characterized mainly by constitutive DNA damage. It is hypothesized that the combination of genotoxic therapies with SIRT6 inhibition may provide a tangible benefit, particularly to poor-prognosis AML patients [141]. It was also shown that patients with *FLT3*-ITD mutation were low expressors of SIRT6. Such correlation was not observed between SIRT6 and *NPM1*, Wilms tumor (*WT1*) mutations, or brain and acute leukemia cytoplasmic (*BAALC)* gene expression. It was further demonstrated that depletion of SIRT6 increased sensitivity to drugs such as idarubicin, AraC, and fludarabine, and confirmed poor EFS and OS in high expressors of SIRT6 [142]. Moreover, Hubner S. et al. reported that high SIRT6 expression had an adverse effect on remission duration in both pediatric and adult AML patients [143].

In another study, Carraway HE et al. investigated DNA hypomethylating agents (HMAs), such as 5-azacytidine and decitabine, both in terms of their effects on reversing DNA methylation and on HDACs actions, including SIRTs. They showed that HMAs increase the enzymatic activity of SIRT6 but have no significant effect on other SIRTs. The authors hypothesized that SIRT6 activation may be an additional HMAs attempt to repair the DNA of leukemic cells and restore tumor-suppressing genes. Thus, the authors suggest that a combination of HMAs and SIRT6 inhibitors or chemotherapeutics may not be effective due to their opposite mechanisms of action [144].

Zhang Y. et al. investigated the effect of LINC00319, a long non-coding RNA, highly expressed in AML, on AML cells. They found that the molecule is responsible for a mechanism that promotes leukemogenesis by increasing SIRT6 expression. Furthermore, the analysis showed that SIRT6 overexpression suppressed the effect of LINC00319 knockdown on AML cells, promoting their growth [145]. Zhang P. et al., however, performed a study in which they proved that knockout of HDAC8 and SIRT6 sensitizes AML cells to NAMPT inhibitors, suggesting their synergistic effect on leukemic blasts [146]. Shi X. et al. conducted a study on NAMPT and nicotinamide nucleotide adenylyltransferase 1 (NMNAT1) and demonstrated that deletion of NMNAT1 reduced nuclear NAD+, leading to weaker SIRT6 and SIRT7 activity. This resulted in reduced deacetylation and p53 activation, thus leading to sensitization of AML to venetoclax. Deletion of SIRT6 and SIRT7 in AML cells also increased apoptosis; interestingly, this effect was not observed in SIRT1 deletion [147]. Furthermore, Yang X. et al. found that DNA (cytosine-5)-methyltransferase 3A (*DNMT3A*) mutation, common in AML, resulted in high NAMPT expression. The authors demonstrated that NAMPT inhibition induced cyclin-dependent kinase complex (cyclin-CDK) depolymerization through SIRT6, which resulted in increased mouse survival and decreased leukemic cell infiltration [148].

SIRT6, a molecule that is important in DNA repair and has an anti-oncogenic effect in most cancers, appears to be sneaky in AML and can work in favor of the disease.

### 4.7. Sirtuin 7

The last member of the SIRT family, SIRT7, acts in both the nucleus and nucleolus, playing a role in multiple processes, like transcription and regulation of metabolism, stress response, aging, tumorigenesis, and genome stability [149]. One of its main biological activities is histone 3 on lysine residue 18 (H3K18) deacetylation, and consequently, chromatin packing, which in turn inhibits transcription [149,150]. Additionally, the desuccinylate ability of SIRT7 has been described, which also alters chromatin and impacts gene expression [151]. In the nucleus, on the other hand, SIRT7 is responsible for stabilizing rDNA heterochromatin, thus influencing the production of ribosomes [152]. In tumorigenesis, SIRT7, unsurprisingly, plays a dualistic role, depending on the biological context. Its crucial genome-stabilizing function counteracts tumor development. Nevertheless, it seems that in cells that are already malignant, SIRT7 can act as either an anti- or pro-oncogenic molecule [153]. Barber MF et al. demonstrated that high H3K18 deacetylation by SIRT7 enhances the persistence of the malignant phenotype of cancer cells [154,155]. In addition, SIRT7 can attenuate transcription of specific tumor-suppressor genes and has the ability to produce rRNA for the metabolic needs of the cancer cell [154,156]. High levels of SIRT7 were associated with aggressive and metastatic prostate and gastric carcinomas [157,158]. Increased expression of SIRT7 has also been described in hepatic, ovarian, breast, and lung cancers, but it appears to be a tumor suppressor in head and neck squamous cell carcinoma [159,160,161,162,163].

In most cases, SIRT7 is well known for its tumor-promoting functions. However, in hematologic malignancies, including AML, the protein seems to act differently. Raza U. et al. described the effect of SIRT7 on nuclear respiratory factor 1 (NRF1) in hematopoietic stem cells, which involves inhibition of transcription of ribosomal proteins, thereby reducing stress and enabling the maintenance of a pool of HSCs, which in turn protects the cells from leukemogenesis [164]. Similar conclusions were reached by Wang Y. et al., who noted that SIRT7 promotes HSCs by inhibiting mitochondrial folding stress and mitochondrial NRF1 activity [165]. These data correlate with results obtained by Nowicki M. et al. in their study, which focused on the efficacy of HSC mobilization prior to autologous transplantation in patients with myeloma and lymphoma. The analysis revealed that upregulation of SIRT7 was noted in patients with high levels of stem cells on day 1 of apheresis. Interestingly, SIRT7 expression was also higher in patients who achieved complete remission before mobilization compared to those who demonstrated partial remission [166]. Kaiser A. et al. showed that SIRT7 expression levels in leukocytes from healthy people decrease in old age. As it is generally known, AML incidence tends to grow with age. Similarly, the authors described low SIRT7 expression in patients with AML and CML. What is more, its expression changed with disease activity—it increased with a positive response to treatment, while progression or relapse was associated with a subsequent decrease in the SIRT7 level [167]. The above data indicate that SIRT7 could serve as a potential biomarker in monitoring treatment response in the above diseases, as well as in the prognosis of HSCs harvesting.

A gene expression analysis in AML patients without cytogenetic abnormalities, performed by Metzeler K. et al., showed that low *SIRT7* mRNA expression was associated with shorter OS, especially in the FLT3-ITD-mutated subgroup (*p* = 0.011) [168]. Our previous preliminary data revealed a reduction in *SIRT7* mRNA levels in patients with *FLT3*-ITD^mut^ (*p* = 0.07) [92]. These results are consistent with data published by Kaiser A. et al., who described higher SIRT7 expression in *FLT3*-ITD-negative patients compared to *FLT3*-ITD^mut^ patients [167]. Furthermore, some key proteins in leukemogenesis, like p53 and NPM1, are substrates for SIRT7 [169,170]. Ianni A. et al. demonstrated that SIRT7 can mediate NPM1 deacetylation, leading in consequence to p53 stabilization under UV-induced genotoxic stress in the skin [171]. On the other hand, Shi H. et al. point to a pathway consisting of NMNAT1-SIRT7-p53 that mediates AML cell survival [147]. Regulatory properties of SIRT7 on p53 levels and function are also described in other studies, but their results are not conclusive [170,172,173,174]. Interestingly, both *SIRT7* and *TP53* are located on chromosome 17, which is affected by frequent aberrations in acute leukemias [23]. These mechanisms need to be further explored, also in the AML context.

The summarized SIRTs role in AML and their interactions are listed in Table 1.

## 5. Therapeutic Applications/Implications

HDAC inhibitors have been evaluated in clinical trials for many hematological malignancies. Vorinostat and belinostat have recently been approved in cutaneous T cell lymphoma (CTCL) and peripheral T cell lymphoma (PTCL), respectively [175,176]. In AML, numerous trials investigating combinations of standard chemotherapy with HDAC inhibitors, including vorinostat (NCT00656617), belinostat (NCT00878722), pabinostat (NCT01451268), entinostat (NCT00313586), and romidepsin (NCT00062075), have been carried out. However, no effective molecule has yet been identified [177]. There are some hopes associated with chidamide, whose synergistic effect with cladribine has been confirmed on AML cell lines (NCT05330364), leading to the cell cycle arrest and apoptosis via the HDAC2/c-Myc/RCC1 (regulator of chromosome condensation 1) axis [178]. Another promising study concerns the combination of belinostat with pevonedistat, a selective NEDD8 inhibitor involved in several mechanisms, including neddylation, which influences cell death (NCT03772925) [4,179,180]. It is worth noting that most of the mentioned molecules inhibit zinc-dependent HDACs of classes I, II, and IV, and therefore do not directly affect SIRTs. Several selective SIRT inhibitors may have applications in AML, such as the SIRT5 inhibitor NRD167 or the SIRT3 inhibitor YC8-0219. Their mechanisms of action are pleiotropic and show activity by modifying metabolic and energetic processes, directing AML cells toward apoptosis, and may also target LSCs survival [86,115]. Some other molecules targeting SIRT, such as Tenovin-6, are the subject of preclinical studies but have not yet been applied in AML [181]. The mechanisms of action of SIRTs and, consequently, their modulators, are still unclear, so taming these molecules for therapeutic purposes poses a challenge and requires further efforts and research. Selected SIRT modulators and their mechanisms of action are listed in Table 2.

## 6. Limitations, Hopes, and Future Challenges

SIRTs are the molecules most commonly associated with the activation of longevity genes, but also the silencing of genes connected with cell aging, whose activity is associated with the development of cardiovascular or neurodegenerative diseases. For this reason, research on SIRTs is mainly concerned with their activation in cardiovascular or neurological disorders, or their impact on lifespan [17,130,212]. In oncology, the issue concerning SIRT appears to be more complex. As mentioned, SIRTs can act in carcinogenesis in two ways by influencing complex epigenetic and biochemical mechanisms. Extension of cell life with SIRTs, as well as modulation of metabolic and energetic processes, can be used to the advantage of cancer [103,140]. On the other hand, some SIRTs in a specific clinical context appear to have tumor-suppressive effects, for instance, by blocking glycolysis, which is required for cell growth [84,97]. These opposing mechanisms described in basic research cause difficulties in finding tangible therapeutic targets.

One study on SIRTs that somehow relates to oncogenesis is a clinical trial on the evaluation of the effect of sirtuin stimulation by resveratrol on the gene expression of proteins that inhibit angiogenic cell apoptosis (NCT05808387). Even though it concerns the assessment of patients with coronary artery disease, it may be a direction for designing studies in oncological diseases, depending on the primary trial results.

However, clinical studies on the use of SIRTs modulators in oncology, even more so in tumors as aggressive and with complex biology as AML, are still lacking.

This review summarizes the knowledge and settles SIRTs in terms of their impact on the development of AML, concluding that most of their activity promotes leukemogenesis. Consequently, this prompts the search for SIRTs inhibitors as therapeutic tools in AML. Despite the promising effects of molecules such as YC8-02, or NRD167, which appear to specifically inhibit AML development at the cellular level, data on their sufficient efficacy and safety are scarce at this point, preventing their clinical application [86,115].

Before clinical trials involving sirtuin modulators in AML can be realistically designed, answers to many questions, also posed in the context of other diseases, need to be known. The activity of SIRTs and the communication with their regulators at the biochemical level, both under physiological conditions and in the tumor microenvironment, need to be explored in greater detail. This knowledge is necessary to try to discover molecules that, in healthy cells, will not interfere with such essential processes as energy metabolism, contrary to cancer cells, where this phenomenon would be desirable. Moreover, a significant obstacle is that the same SIRT can act pro-oncogenically in the context of certain cancers and suppressively against others. Additionally, by inhibiting the activity of SIRTs to act proapoptotically, there is a risk of affecting healthy cells, accelerating their aging, and resulting in long-term complications that are tough to predict. Indeed, modulation of the activity of such proteins may mean that by acting therapeutically on one disease process, we increase the risk of developing another. The final step, but also a challenging one, is the introduction of AML treatment based on regulating SIRT activity into everyday clinical practice. The complexity of the mechanisms responsible for the etiopathogenesis of this disease, as well as the frequent acquisition of resistance to the treatment lines used, is a current problem for every drug known to date.

The hope is that with better and more common research tools, as well as expanding knowledge of AML at the molecular level, there is an opportunity to look for drug combinations in specific patient subgroups that may improve their prognosis. For example, patients refractory to venetoclax may potentially benefit from SIRT5 inhibition [119]. Another example is the sensitization of LSCs to venetoclax through SIRT3 inhibition, which may be relevant in relapsed AML [87]. An unquestionable step forward would be the possibility of specifically inhibiting SIRT1, which is responsible for p53 silencing [43]. On the other hand, it is difficult to predict the consequences of such a strategy given the impact on processes such as energy metabolism, calorie restriction, oxidative stress, inflammation, etc. [213]. Another great challenge is to determine which SIRTs are specifically promoters and which are suppressors of leukemogenesis, which may not be possible given the heterogeneity of AML and the dynamics of changes occurring in its microenvironment.

Despite extensive fundamental research and deepening knowledge of SIRTs in oncogenesis, the above limitations continue to hinder the invention of effective SIRTs-based therapies in AML. However, the multiplicity of the mechanisms these molecules are involved in, from epigenetic to biochemical and cellular levels, offers hope that taming them in the future will translate into tangible improvements in the prognosis of AML patients.

## 7. Conclusions

Our knowledge of SIRT regulation mechanisms is constantly expanding, and it may result in new diagnostic and therapeutic possibilities. Despite promising results of preclinical studies, we still have not identified effective selective SIRT modulators in clinical trials. The potential reason might be the dual nature of SIRTs in the context of cancer, which makes the development of anti-cancer drugs challenging. Nevertheless, a better understanding of the role of SIRTs in AML, and particularly in the regulation of p53, may translate into a selection of novel targets for a more personalized therapeutic approach. Undoubtedly, SIRTs play a complex role in AML, and, especially for certain AML subtypes, interference in these pathogenetic pathways may represent a treatment strategy.

## Figures and Tables

**Figure 1 cancers-17-01009-f001:**
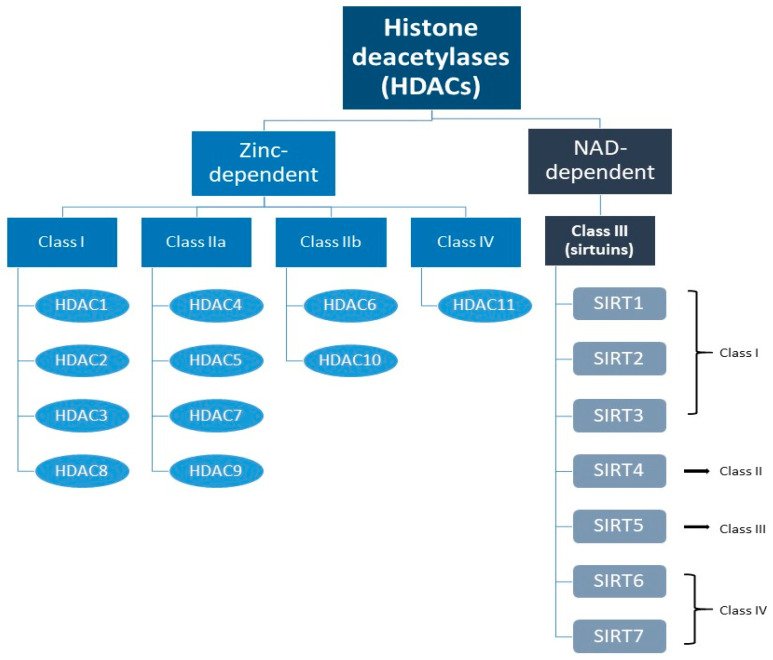
Division of histone deacetylases with particular reference to sirtuins (SIRTs) and their classes.

**Figure 2 cancers-17-01009-f002:**
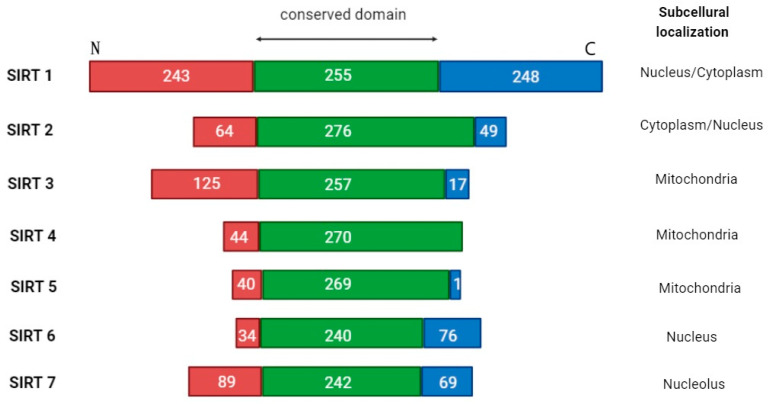
The overall architecture of SIRT1-7 domains. Human sirtuins have conserved catalytic cores (green center) and variable N- (red end) and C- (blue end) terminals. For each sirtuin isoform, the boundaries of the catalytic core were determined using UniProt. Differences in the cellular localization of each sirtuin are highlighted.

**Table 1 cancers-17-01009-t001:** Roles of sirtuins (SIRTs) in AML.

Sirtuin (SIRT)	Class	Main Cell Localization	Potential Signaling Pathways Impact and Affected Molecules	Functions	Level	Probable Role in AML	Ref.
SIRT1	I	Nucleus Cytoplasm	p53, c-Myc, USP22, FOXO1, FOXP1, STAT5, DOT1L, miR-9	Deacetylase Deacylase	High	Promoter	[43,44,47,55,56,57,58,60]
SIRT2	I	Cytoplasm Nucleus (during mitosis)	NAMPT, MAPK, VEGF, MRP1, ERK1/2, BCL-2, p53, Caspase-1, AKT/GSK3β/β-catenin, PI3K/AKT/mTOR, pentose-phosphate cycle, p21, cyclin E1, miR-140-5p, miR-145	Deacetylase Deacylase	High	Promoter	[68,69,73,74,75,76,77,78,79,80]
SIRT3	I	Mitochondria	PTEN, MDM2, p53, cyclophilin D, IDH2, tricarboxylic acid cycle, Fatty acid oxidation, sumoylation, HES1, Notch1	Deacetylase Decrotonylase	Low	Undetermined/ Bi-directional	[83,84,85,86,89,90]
SIRT4	II	Mitochondria	Glutamine metabolism	Deacetylase Deacylase ADP-rybosil-transferase,	Low (high according to some sources)	Undetermined/ Bi-directional	[94,96,97]
SIRT5	III	Mitochondria	BAX/BCL-2, oxidative phosphorylation	Deacetylase Demalonylase Deglutarylase Desuccinylase	No data	Promoter	[104,105,106,107,110,111,113,114]
SIRT6	IV	Nucleus	NAMPT, NMNAT1, cyclin-CDK	Deacetylase Deacylase ADP-rybosil-transferase	High	Promoter	[140,146,147,148]
SIRT7	IV	Nucleus Nucleolus	NRF1, NPM1, p53	Deacetylase Dessuccinylase	Low	Suppressor	[147,151,164,165,167,171]

AML—acute myeloid leukemia, c-Myc—cellular myelocytomatosis oncogene, USP22—ubiquitin-specific peptidase 22, FOXO1—forkhead box protein O1, FOXP1—forkhead box protein 1, STAT5—signal transducer and activator of transcription 5, DOT1L—disruptor of telomeric silencing 1-like, miR-9—microRNA-9, NAMPT—nicotinamide phosphoribosyltransferase, MAPK—mitogen-activated protein kinase, VEGF—vascular endothelial growth factor, MRP1—multidrug resistance-associated protein 1, ERK1/2—extracellular signal-regulated kinase 1 and 2, BCL-2—B-cell lymphoma 2 protein, AKT—protein kinase B, GSK3β—glycogen synthase kinase-3 beta, PI3K—phosphoinositide-3-kinase, mTOR—mammalian target of rapamycin, miR-140-5p—microRNA-140-5p, miR-145—microRNA-145, PTEN—phosphatase and tensin homologue deleted on chromosome 10 (PTEN), MDM2—mouse double minute 2 homolog, IDH2—isocitrate dehydrogenase (NADP(+)) 2, HES—hairy and enhancer of split-1, Notch1—neurogenic locus notch homolog protein 1, BAX—bcl-2-like protein 4, NMNAT1—nicotinamide nucleotide adenylyltransferase 1, cyclin-CDK—cyclin-dependent kinase complex, NRF1—nuclear respiratory factor 1, NPM1—nucleophosmin 1.

**Table 2 cancers-17-01009-t002:** Selected sirtuin (SIRT) modulators.

Molecule	Mechanism of Action	Targeted Sirtuin (SIRT)	Additional Mechanisms	Reference
Selisistat (EX-527)	Inhibitor	SIRT1		[182]
Nicotinamide-d_4_ Nicotinamide-^13^C_6_ Nicotinamide-^15^N,^13^C_3_	Inhibitors	SIRT1	- NAD+ redox homeostasis	[183,184,185]
Z26395438 (compound **1**)	Inhibitor	SIRT1		[186]
Antiproliferative agent-17	Inhibitor	SIRT1	- Anticancer activity - Gram+ bacteria inhibition	[187]
Sirt1/2-IN-3 (compound **PS9**)	Inhibitor	SIRT1 SIRT2	- p53 deacetylation blockade	[188]
Sirt1/2-IN-2 (compound **hsa55**)	Inhibitor	SIRT1 SIRT2	- p53 deacetylation blockade	[188]
Tenovin-1	Inhibitor	SIRT1 SIRT2	- p53 activation - Dihydroorotate dehydrogenase inhibitor	[5,189,190,191]
Tenovin-6	Inhibitor	SIRT1 SIRT2	- p53 activation - Increase in FLT3-ITD+ AML LSCs ablation	[181]
4′-Bromo-resveratrol	Inhibitor	SIRT1 SIRT3	- Mitochondrial metabolic reprogramming	[192]
Sirt1/2-IN-4 (compound **PS3**)	Inhibitor	SIRT1 SIRT2 SIRT3	- Potential anticancer activity - p53 deacetylation blockade	[188]
BZD9Q1	Inhibitor	SIRT1 SIRT2 SIRT3	- Apoptosis and necrosis induction, G_2_/M cycle arrest	[193]
SIRT1/2/3-IN-1 (compound **10**)	Inhibitor	SIRT1 SIRT2 SIRT3	- Potential anticancer activity	[194]
Resveratrol	Activator	SIRT1		[195]
SIRT1 activator 1(compound **3**)	Activator	SIRT1		[196]
JFD00244	Inhibitor	SIRT2	- Nsp-16 inhibitor	[197,198]
SirReal1, SirReal2	Inhibitors	SIRT2		[199]
3-aryl-mercapto-butyrylated peptide derivative	Inhibitor	SIRT2		[200]
SIRT2-IN-12 (compound **3**) (xanthone derivative)	Inhibitor	SIRT2		[201]
Mz325	Inhibitor	SIRT2	- HDAC inhibitor	[202]
HSP70/SIRT2-IN-2 (Compounds **1a**)	Inhibitor	SIRT2	- HSP70 inhibitor - Anticancer activity	[203]
SIRT2/6-IN-1 (Compound **5**)	Inhibitor	SIRT2 SIRT6	- Increase in H3K9 acetylation - Increase in glucose uptake - Reduction in TNF-alpha secretion in cells	[204]
YC8-02	Inhibitor	SIRT3	- Decrease in ATP levels - Reduction in fatty acid oxidation activity - LSC targeting	[86]
SIRT4-IN-1 (compound **69**)	Inhibitor	SIRT4		[205]
NRD167	Inhibitor	SIRT5	- Increase in mitochondrial superoxide - Reduction in oxidative phosphorylation - AML cells apoptosis induction	[115]
179MC3482	Inhibitor	SIRT5		[206]
SIRT5 inhibitor 1	Inhibitor	SIRT5		[207]
SIRT5 inhibitor 8 (compound **10**)	Inhibitor	SIRT5	- Anticancer potential	[208]
SIRT5 inhibitor 9 (compound **14**)	Inhibitor	SIRT5	- Anticancer potential	[208]
SIRT5 Inhibitor 6 (2,4,5-trisubstituted pyrimidine derivative)	Inhibitor	SIRT5	- Modulating sepsis - AKI	[209]
SIRT5 inhibitor 7 (compound **58**) (2,4,5-trisubstituted pyrimidine derivative)	Inhibitor	SIRT5	- Anti-inflammatory activity	[209]
SIRT6-IN-2 (compound **5**)	Inhibitor	SIRT6	- Increase in H3K9 acetylation - Increase in glucose uptake in cultured cells	[210]
UBCS039	Activator	SIRT6	- Anti-inflammatory response - Oxidative stress alleviation	[193,211]

NAD+—nicotinamide adenine dinucleotide, FLT3-ITD—internal tandem duplication in the FMS-like tyrosine kinase 3 gene, AML—acute myeloid leukemia, LSCs—leukemic stem cells, Nsp-16—non-structural protein 16, HDAC—histone deacetylases, HSP70—70 kilodalton heat shock proteins, H3K9—9th lysine residue of the histone H3 protein, TNF-alpha—tumor necrosis factor-alpha, ATP—adenosine triphosphate, AKI—acute kidney injury.

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
