# Peer review of "The Role of the Sirtuin Family Histone Deacetylases in Acute Myeloid Leukemia—A Promising Road Ahead"

_cancers, 2025, doi:10.3390/cancers17061009_

Round 1

Reviewer 1 Report

Comments and Suggestions for Authors

The authors have compiled a comprehensive collection of information on Sirtuins in AML. I have a few suggestions

  1. Since the authors focus exclusively on AML, they should provide more data on its prevalence and geographical distribution in line 46.
  2. The values in lines 101–102 should be rechecked against the latest data and cited properly.
  3. The authors should include a protein or domain-based figure of Sirtuins (SIRT1 to SIRT7) to highlight the major differences between them.
  4. After reading the Therapeutic Applications/Implications section, I found it somewhat unclear and confusing. The authors should more clearly specify which molecule or inhibitor acts at which stage or condition of AML. Additionally, they must provide data on the percentage of inhibition or the efficacy of each inhibitor in preventing AML progression in Table 2.
  5. Although the authors have mentioned some limitations in few sections, I believe the authors may make a section on limitations and future challenges for better reading and clarity for readers.

Author Response

Dear Sir/Madam,

Thank you sincerely for all your comments and corrections. We have incorporated all suggestions into the revised manuscript and highlighted the changes in colour. We are convinced that this has made the article much more valuable. We have included below our responses to the individual comments.

Yours faithfully,
The authors

1. Since the authors focus exclusively on AML, they should provide more data on its prevalence and geographical distribution in line 46.

Response: Thank you for pointing this out. We have added a few sentences about the prevalence, geographic distribution and general introduction to AML in lines 45-53.

2.The values in lines 101–102 should be rechecked against the latest data and cited properly.

Response: We have corrected the data according to newest literature and cited properly, thank you for this comment (lines 109-111).

3.The authors should include a protein or domain-based figure of Sirtuins (SIRT1 to SIRT7) to highlight the major differences between them.

Response: We have created and added a domain-based figure of SIRT1-7, thank you.

4. After reading the Therapeutic Applications/Implications section, I found it somewhat unclear and confusing. The authors should more clearly specify which molecule or inhibitor acts at which stage or condition of AML. Additionally, they must provide data on the percentage of inhibition or the efficacy of each inhibitor in preventing AML progression in Table 2.

Response: Thank you for drawing attention to this clinically important issue. Unfortunately, at this point, there is no evidence that HDAC modulators, or Sirtuin modulators in particular, are applicable to the treatment of AML. The studies we refer to are in preclinical stages, and the efficacy of these molecules has not been confirmed in clinical trials in AML. The section on therapeutic implications is intended to alert readers to the still unexplored field of the use of sirtuin modulators and other HDACs in the treatment of AML, and to inspire further exploration. Instead, we have added a few sentences in this section describing the potential mechanisms of action of HDAC modulators in AML, which I hope will enrich the chapter with a broader view. In accordance with the above, it is also not possible to estimate response rates in AML for the individual molecules listed in Table 2. We have revised the table to make it more relevant and readable, though. We find all the suggestions pertinent and appreciate them.

5. Although the authors have mentioned some limitations in few sections, I believe the authors may make a section on limitations and future challenges for better reading and clarity for readers.

Response: We considered this a very good point and proceeded to create a separate section on limitations and future challenges. It summarizes and covers the topic of difficulties that still need to be faced. Thank you very much.

Reviewer 2 Report

Comments and Suggestions for Authors

The review article by Strzalka et al focuses on the role of sirtuins in AML. My comments are noted below.

  1. Currently there are no SIRT in clinical trials for AML. Please further elaborate this in the conclusion section of the review article.

Could the lack of SIRT in clinical trials for AML be due to the complex role of sirtuins, including the potential for off-target effects and concerns about impacting normal hematopoietic stem cells, which could lead to significant side effects. Nevertheless, recent studies highlight SIRT5 as a promising target due to its specific role in AML cell survival without significantly affecting healthy cells as noted in ref 117.

  1. Line 304 mentioned “O’Brien et al. demonstrated that SIRT3 inhibition reduces fatty acid oxidation activity and substrate availability for the tricarboxylic acid cycle, thereby affecting oxidative phosphorylation, decreasing ATP production, and leading to LSC death”. I presume this SIRT3 inhibitor is YC8-02. Please include YC8-02 this in Table 2.

  1. Please revise table 2 to indicate which SIRT inhibitors have promising progress in targeting AML. For example,

SirReal2 for targeting SIRT2

YC8-0219 for targeting SIRT3

NRD167 for targeting SIRT5

Author Response

Dear Sir/Madam,

Thank you sincerely for all your comments and corrections. We have incorporated all suggestions into the revised manuscript and highlighted the changes in colour. We are convinced that this has made the article much more valuable. We have included below our responses to the individual comments.

Yours faithfully,

The authors

1. Currently there are no SIRT in clinical trials for AML. Please further elaborate this in the conclusion section of the review article.

Could the lack of SIRT in clinical trials for AML be due to the complex role of sirtuins, including the potential for off-target effects and concerns about impacting normal hematopoietic stem cells, which could lead to significant side effects. Nevertheless, recent studies highlight SIRT5 as a promising target due to its specific role in AML cell survival without significantly affecting healthy cells as noted in ref 117.

Response: We considered this a very good point and proceeded to create a separate section on limitations and future challenges to conclude the topic. It summarizes and covers the difficulties that still need to be faced. Thank you very much.

2. Line 304 mentioned “O’Brien et al. demonstrated that SIRT3 inhibition reduces fatty acid oxidation activity and substrate availability for the tricarboxylic acid cycle, thereby affecting oxidative phosphorylation, decreasing ATP production, and leading to LSC death”. I presume this SIRT3 inhibitor is YC8-02. Please include YC8-02 this in Table 2.

Response: Yes, indeed. We have included YC8-02 both in the text and in Table 2. Thank you for your comment.

3. Please revise table 2 to indicate which SIRT inhibitors have promising progress in targeting AML. For example,

SirReal2 for targeting SIRT2

YC8-0219 for targeting SIRT3

NRD167 for targeting SIRT5

Response: We have added these and several other modulators to the table and highlighted their effects in the context of AML. Thank you very much.

Reviewer 3 Report

Comments and Suggestions for Authors

This is a very thorough and comprehensive review of the role of sirtuin family histone deacetyalses in AML, with a very detailed description of their background and their potential role in the biology of AML. The large number of references included by the authors is impressive, highlighting the amount of work that has gone into this manuscript. It is clear and relatively easy to follow with the description of the sirtuin family, but the manuscript can benefit from a more concise summary of the different SIRTs, in their potential contribution to AML and its therapy which remains suboptimal, and what the current limitations and barriers are. 

The manuscript is interesting, but readers would find the section on therapeutic applications and limitations more useful if the authors can expand on this section to make the knowledge of the sirtuins more clinically relevant. Table 2 on selected SIRT modulators is very clear.

Overall, an interesting review which is well written.

Author Response

Dear Sir/Madam,

Thank you sincerely for all your comments and corrections. We have incorporated all suggestions into the revised manuscript and highlighted the changes in colour. We are convinced that this has made the article much more valuable. We have included below our response to the individual comments.

Yours faithfully,

The authors

This is a very thorough and comprehensive review of the role of sirtuin family histone deacetyalses in AML, with a very detailed description of their background and their potential role in the biology of AML. The large number of references included by the authors is impressive, highlighting the amount of work that has gone into this manuscript. It is clear and relatively easy to follow with the description of the sirtuin family, but the manuscript can benefit from a more concise summary of the different SIRTs, in their potential contribution to AML and its therapy which remains suboptimal, and what the current limitations and barriers are. 

The manuscript is interesting, but readers would find the section on therapeutic applications and limitations more useful if the authors can expand on this section to make the knowledge of the sirtuins more clinically relevant. Table 2 on selected SIRT modulators is very clear.

Overall, an interesting review which is well written.

Response: Thank you very much for your comment and suggestions. We considered this a very good point and proceeded to create a separate section on limitations and future challenges. It summarizes and covers the topic of difficulties that still need to be faced. Thank you very much for your support, we appreciate it.

Round 2

Reviewer 1 Report

Comments and Suggestions for Authors

The authors provided a satisfactory response.